# 7-Isopentenyloxycoumarin: What Is New across the Last Decade

**DOI:** 10.3390/molecules25245923

**Published:** 2020-12-14

**Authors:** Francesca Preziuso, Salvatore Genovese, Lorenzo Marchetti, Majid Sharifi-Rad, Lucia Palumbo, Francesco Epifano, Serena Fiorito

**Affiliations:** 1Department of Pharmacy, University “G. d’Annunzio” of Chieti-Pescara, Via dei Vestini 31, 66100 Chieti Scalo (CH), Italy; francesca.preziuso@unich.it (F.P.); s.genovese@unich.it (S.G.); marchetti_lorenzo@libero.it (L.M.); lucia.palumbo@unich.it (L.P.); serena.fiorito@unich.it (S.F.); 2Department of Range and Watershed Management, Faculty of Water and Soil, University of Zabol, Zabol 98613-35856, Iran; majidsharifirad@uoz.ac.ir

**Keywords:** Apiaceae, *Citrus*, coumarins, 7-isopentenyloxycoumarin, prenylation, Rutaceae

## Abstract

7-Isopentenyloxycoumarin is among the most widespread naturally occurring prenyloxy umbelliferone derivatives. This secondary metabolite of mixed biosynthetic origin has been typically isolated from plants belonging to several genera of the Rutaceae and Apiaceae families, comprising widely used medicinal plants and in general plants with beneficial effects on human welfare, as well as edible fruits and vegetables. Although known for quite a long time (more than 50 years), only in the last two decades has this natural compound been revealed to exert powerful and promising pharmacological properties, such as active cancer chemopreventive, antibacterial, antiprotozoal, antifungal, anti-inflammatory, neuroprotective, and antioxidant properties, among the activities best outlined in the recent literature. The aim of this comprehensive miniature review article is to detail the novel natural sources and the effects described during the last decade for 7-isopentenyloxycoumarin and what has been reported on the mechanisms of action underlying the observed biological activities of this oxyprenylated secondary metabolite. In view of the herein described data, suggestions on how to address future research on the abovementioned natural product and structurally related derivatives in the best ways according to the authors will be also provided.

## 1. Introduction

Coumarins represent a wide group of naturally occurring compounds, mainly found in the families of Apiaceae, Asteraceae, Leguminosae, Moraceae, Oleaceae, Rutaceae, and Thymelaeaceae [1]. More than 1300 chemical entities have been characterized to date, but most chemical and pharmacological studies have been accomplished on either coumarin itself or structurally simple natural and semisynthetic derivatives. Coumarins can be divided into four categories: (a) substituted coumarins, (b) ring-fused coumarins, (c) *C-*prenylcoumarins, and (d) *O-*prenylcoumarins. In particular, these two latter groups comprise phytochemicals in which isoprenyl and/or terpenyl side chains are attached to the benzopyrone ring either directly or through one or more phenoxy moieties via an ether bond. However, while *C-*prenylcoumarins have been investigated in detail from chemical, phytochemical, and pharmacological points of view [2], prenyloxycoumarins, considered for many years as mere biosynthetic intermediates of linear, furano- and pyranocoumarins, have only in the last decade been characterized as secondary metabolites exerting valuable biological activities [3]. Very recently the properties of notable examples in this group of oxyprenylated phenylpropanoids, namely auraptene (7-geranyloxycoumarin) and umbelliprenin (7-farnesyloxycoumarin), outlined in the literature during the last decade have been reviewed [4]. The aim of this miniature review is to detail from phytochemical and pharmacological points of view, the features of an additional biologically active coumarin derivative, 7-isopentenyloxycoumarin (Figure 1)

Data collected in this article were taken from main Internet databases such as Scopus, Web of Science, Pubmed, and Google Scholar using the corresponding keyword as the bibliographic search parameter. Analysis of the literature indicated that, since the publication of the last review article on the same topic in 2009 by Epifano and coworkers [5], 19 research articles have appeared. Data outlined herein will also be tentative suggestions to the overall scientific community working in the field on how to stimulate and address future research activities to study and employ oxyprenylated coumarins in general in the best ways.

## 2. New Natural Sources

Like several other oxyprenylated coumarins, 7-isopentenyloxycoumarin has also been found in a very restricted number of plant families, namely Apiaceae, Asteraceae, and Rutaceae, as previously reported [5]. Its biosynthetic pathway, one of the very few for oxyprenylated phenylpropanoids characterized until now, was discovered in cell suspension cultures of *Ammi majus* L. (Apiaceae) by Hamerski and coworkers in 1990 [6]. Thus, 7-isopentenyloxycoumarin is obtained by the coupling reaction between umbelliferone and dimethyallyl diphosphate catalyzed by an enzyme located in the endoplasmic reticulum of plant cells, named dimethylallyl diphosphate umbelliferone transferase (DDU-7 transferase E.C. 2.5.1) (Figure 2).

Supplementation with the unprenylated coumarin umbelliferone to dill cells cultured in a medium containing Murashige and Skoog base medium with added saccharose (30 g/L) and 2,4-dichlorophenoxyacetic acid (1.13 μM), elicits the biosynthesis of 7-isopentenyloxycoumarin [7].

A straightforward biomimetic synthesis of 7-isopentenyloxycoumarin is also available and consists of a single step of a Williamson’s reaction between umbelliferone and 3,3-dimethylallyl bromide in refluxing acetone promoted by dry K_2_CO_3_ as the base [8].

First isolated in 1966 by Prokopenko from the fruits of *Libanotis intermedia* Rupr. (common name “moon carrot”, syn. *Seseli libanotis* (L.) W. D. J. Koch, Apiaceae) [9], 7-isopentenyloxycoumarin was believed until a few years ago to be part of the phytochemical pool of a very restricted group of plant families (e.g., Apiaceae, Asteraceae, Rutaceae) [6]. Recent investigations carried out during the last 5 years did widen the number of species and natural sources in general from which the title oxyprenylated coumarin can be found.

In 2016, Taddeo and coworkers demonstrated for the first time in the literature the occurrence of 7-isopentenyloxycoumarin from something not belonging to the vegetable kingdom [10]. In their investigation devoted to comparing the efficiency of different extraction methodologies of selected prenylated and unprenylated coumarins and cinnamic acid derivatives from dry raw Italian propolis followed by quantification by HPLC analysis, it was shown that 7-isopentenyloxycoumarin could be easily extracted with good yields from this food matrix using several techniques such as maceration with pure ethanol (126.68 µg/g of dry propolis), ultrasonication with a 7:3 ethanol/water mixture (181.29 µg/g), and in the presence of β-cyclodextrin (β-CD) as an auxiliary agent either in water (1.5 *w/w*, 172.35 µg/g) or by far most efficiently in olive oil (295.84 µg/g) as the solvent. For raw propolis the latter has been seen as the most effective extraction route. The better extraction yields recorded with β-CD may be explained by the tight interaction of the 3,3-dimethylallyoxy side chain with the lipophilic environment represented by the inner cavity of the cyclic polysaccharide, as already evidenced by Tanaka and coworkers in 2010 [11]. Enforcing this hypothesis is the experimental knowledge that umbelliferon, the unprenylated parent coumarin of 7-isopentenyloxycoumarin, has been extracted in very low quantities under the same experimental conditions, while maceration with ethanol provides nearly equal concentration values. This study also indicated for the first time that a bee product like propolis could be regarded as an additional natural source of 7-isopentenyloxycoumarin and biologically active oxyprenylated phenylpropanoids in general. It suggested that the beneficial effects of propolis for human health can be also ascribed to less abundant phytochemicals, like those just mentioned, that could biologically act in synergy with major products typically found in great quantities in propolis like flavonoids and phenolic acids. For example, the reported neuroprotective properties of propolis [12] could be also ascribed to the presence of 7-isopentenyloxycoumarin, known to exert such effects both in vitro and in vivo [13,14]. The paper by Taddeo and coworkers can also be considered as a great stimulus to continue the search of such secondary metabolites in other bee-derived preparations such ashoney, pollen, royal jelly, and bee wax. Indeed, this kind of investigation would be unprecedented in the literature in widening the current knowledge on the chemical composition of healthy bee products.

The same research group adopted a similar approach in comparing several extraction methodologies of the above cited prenylated and unprenylated phytochemicals in analyzing by HPLC the chemical composition of three plants belonging to the Apiaceae family, namely dill (*Anethum graveolens* L.), anise (*Pimpinella anisum* L.), and wild celery (*Angelica archangelica* L.), which all represent notable examples of food and medicinal plants [15]. In this case Taddeo and coworkers revealed how 7-isopentenyloxycoumarin was not part of the secondary metabolite pool of *A. graveolens*, and was detected in very small amounts in extracts from *P. anisum* (3.34 µg/g and 1.08 µg/g of dry extract after maceration with water containing β-CD 1.5% *w/w* and ultrasonication using this same mixture). It was by far most abundant in *A. archangelica*, reaching the best values after maceration (204.76 µg/g) and ultrasonication (177.60 µg/g) in ethanol.

In the same way, 7-isopentenyloxycoumarin was determined as a component of the flower extract of *Amaranthus retroflexus* L. (Amaranthaceae), a herb typically used as a food in central and southern Italy and several other geographic areas of the world. The concentration value recorded was 12.36 µg/g of dry extract after a 96 h maceration with acetone [16]. This last investigation is of particular importance as it deals with the characterization of the title compounds for the first time from a plant species belonging to the Amaranthaceae family, for which the presence of oxyprenylated secondary metabolites was unknown until 2017.

In the same year, Lee and coworkers characterized the presence of 7-isopentenyloxycoumarin in five *Isocoma* spp. (Asteraceae) native to southwestern United Stated and northern Mexico, namely *Isocoma pluriflora* (Torr. and A. Gray) Greene, commonly known as “southern goldenbush”, *Isocoma tenuisecta* Greene, “burrow goldenweed”, *Isocoma azteca* G. L. Nesom, “aztech goldenbush”, *Isocoma acradenia* (Greene) Greene, “alkali goldenbush”, and finally *Isocoma rusbyi* Greene, “Rusby’s goldenbush”. All these plants are poisonous and cause trembles in livestock, although it may be that such a toxicity does not relate to the presence of the title compound. 7-Isopentenyloxycoumarin was detected in amounts ranging from 0.07 µg/mg of dry extract to 1.1 µg/mg and was not found in *I. pluriflora* and *I. azteca* [17]. Furthermore, for the *Isocoma* genus, the one described by Lee and coworkers is the first report on the isolation and structural characterization of an oxyprenylated coumarin.

The interest in the characterization of 7-isopentenyloxycoumarin in edible and medicinal plants greatly increased over the next three years. In 2018 the content of this secondary metabolite in oils used as food or as beneficial remedies was reported. In the first manuscript, Scotti and coworkers set up an efficient extraction procedure for tea tree oil (TTO), the well-known and fashionable essential oil obtained by the stem distillation of leaves and branches of *Melaleuca alternifolia* (Maiden and Betch) Cheel, a plant native to Australia and belonging to the Myrtaceae family [18]. An experimental procedure, with the aim of separatingin the solid phase the highest quantities of oxyprenylated phenylpropanoids including 7-isopentenyloxycoumarin, was applied. Such a procedure consisted of the partition of TTO between a saturated NaHCO_3_ aqueous solution and *n*-hexane, followed by its adsorption on solid matrices represented by silica gel(pre-treated with Et_3_N) and alumina (Brockmann activity II). The latter, coupled to desorption with CH_2_Cl_2_, proved to be the most efficient process and provided 7-isopentenyloxycoumarin in a concentration of 44.03 µg/g of TTO, more than 2-fold in respect to the “classic” overnight maceration in *n*-hexane. It is also noteworthy that the methodology set up by Scotti and coworkers also allowed a complete deterpenation of TTO leading to a selective extraction of oxyprenylated secondary metabolites. Furthermore, in this case, the experiment just described is the first example in the literature of the presence of 7-isopentenyloxycoumarin in a species belonging to the Myrtaceae family.

In the same year, Ferrone and coworkers tested the capacity of deep eutectic solvent dispersive liquid–liquid micro-extraction (DES-DLLME) as a novel and alternative extraction procedure for oxyprenylated secondary metabolites using edible oils (e.g., olive, soy, peanuts, corn, and sunflower) as the matrices [19]. Among the four ionic liquids employed in the course of this investigation, namely glycolic acid/trimethylglicyne, 2-furoic acid/trimethylglicyne, *S*-(+)-mandelic acid/trimethylglicyne, and phenylacetic acid/trimethylglicyne, the latter showed the best performance in terms of higher recoveries and enrichment factor and was thus selected to further perform extractions for the UHPLC/PDA quantification of 7-isopentenyloxyxcoumarin in the oils cited above. This secondary metabolite was detected in all matrices except for sunflower oil. Concentration values were 1.82 μg/mL, 12.92 μg/mL, 1.20 μg/mL, and 2.73 μg/mL for olive, soy, peanuts, and corn oils, respectively. Although parent umbelliferone was recorded as the most abundant coumarin, the findings described in the manuscript by Ferrone and coworkers clearly indicate how prenylation should be considered as an effective metabolic step for the biosynthetic schemes of plants from which the respective oils are obtained, and represents the first example of isolation of the title active principle from plants belonging to the Oleaceae and Poaceae families.

One year later the same research group assessed the presence of 7-isopentenyloxycoumarin in common foods such as spinach (leaves), goji (berries), and quinoa (seeds), applying different mixtures of extracting solvents, namely EtOH, H_2_O/EtOH 3:7, and H_2_O/EtOH 7:3, under microwave irradiation coupled to UHPLC/PDA analysis [20]. In this case ethanol was shown to be the most efficient in terms of extractive yields. 7-Isopentenyloxycoumarin was detected only in extracts from spinach leaves and goji berries in concentration values of 2.33 µg/g and 8.37 µg/g of dry extract, respectively. Noteworthy results from this study were the confirmation of the presence of an oxyprenylated coumarin in the Amaranthaceae family after those obtained with *A. retroflexus* (in the case of *Spinacia oleracea* L.) and the first report of its determination in a species belonging to the Solanaceae family (in the case of *Lycium barbarum* L.).

The most recent articles in the literature concern the application of a novel extraction procedure based on the use of subcritical butane. Although this method was applied to already known sources of 7-isopentenyloxycoumarin like *Citrus paradisi* L. (Rutaceae) [21] and *Artemisia vulgaris* (Asteraceae) [22], both reports illustrate the great potential for the future of subcritical butane as an extraction solvent. In the case of grapefruit seeds, quantities determined for 7-isopentenyloxycoumarin (234.00 µg/g of dry extract) were higher than those obtained by any other extraction methodologies employed so far.

The novel natural sources from which 7-isopentenyloxycoumarin has been determined are listed in the Table 1, with known sources highlighted in bold.

In this context the studies depicted above suggest to the overall scientific community to take into great consideration oxyprenylated phenylpropanoids, like 7-isopentenyloxycoumarin, among the plethora of biologically active compounds nowadays available from the plant kingdom and nature in general, and to identify novel and alternative sources of this rare but valuable and promising class of secondary metabolites.

## 3. Biological Activity

### 3.1. Cytotoxic Activity

During the last decade, the pharmacological profile of 7-isopentenyloxycoumarin was further and better characterized in particular concerning its antitumor properties.

In this context, Bruyere and coworkers tested this oxyprenylated coumarin in vitro as growth inhibitory agents of a panel composed of six human cancer cell lines and found that it was weakly active on OE21 (human esophageal carcinoma) and LoVo (human colorectal adenocarcinoma) cell lines with IC_50_ values of 79 µM and 82 µM, respectively [23]. Two years later, in 2013, Valiahdi and coworkers investigated the cytotoxic effects of individual phytochemicals isolated from *Ferula* spp. against a panel of three human cancer cell lines, namely ovarian carcinoma (CH1), lung cancer (A549), and melanoma (SK-MEL-28) using the (3-(4,5-dimethylthiazol-2-yl)-2,5 diphenyltetrazolium b bromide) (MTT) assay [24]. 7-Isopentenyloxycoumarin however was found to exert only a very modest effect with an IC_50_ value > 250 µM. One year later, Haghighi and coworkers assayed the cytotoxic effect of this secondary metabolite on 5637 cells, a transitional cell carcinoma cell line (bladder cancer) using the same test with human dermal fibroblast (HDF) as the “normal” cell line [25]. The MTT assay was performed in both cell lines to record the cytotoxic activity of 7-isopentenyloxycoumarin in the concentration range 10–100 µg/mL for three consecutive days. This oxyprenylated coumarin exhibited a time- and dose-dependent effect. IC_50_ values for 5637 cells were 76, 76, and 65 μg/mL after 24, 48, and 72 h of treatments, respectively. In the same concentration range and times, 7-isopentenyloxycoumarin did not show an appreciable cytotoxic effect against HDF cells. Gaining further insights into its mechanism of action, microscopic observation on 5637 cells revealed an extensive cytoplasmic granulation and cell death. 4′,6-Diamidin-2-phenylindole (DAPI) staining and alkaline comet assay showed that at a concentration of 65 μg/mL, a high condensation of chromatin and nuclear fragmentation occurred in more than 89% of treated cancer cells. Furthermore, 43% of the latter exhibited DNA damage, while HDF cells were practically not affected. The increase of caspase 3 activity was about 2.5-fold higher in cancer cells in respect to normal ones, suggesting that 7-isopentenyloxycoumarin could behave as a proapoptotic agent. Finally, at the same concentration mentioned above, 7-isopentenyloxycoumarin induced an arrest of the cycle in 5637 cells in the G2/M phase after 24 h treatment. In the same year, Rezaee and coworkers investigated the antigenotoxic properties of this phytochemical on human lymphocytes under oxidative stress conditions using the comet assay [26]. 7-Isopentenyloxycoumarin did not show appreciable effects on cell viability even at high concentrations. Isolated lymphocytes were treated with H_2_O_2_ at a concentration of 25 µM and were found to induce 45% DNA damage. The effect of the oxyprenylated coumarin was revealed by its incubation with cells with and without H_2_O_2_ in the concentration range 10–200 μM. Under these conditions, 7-isopentenyloxycoumarin exhibited a dose-dependent reduction in DNA damage (0.88–8%). Quite interestingly, the parent coumarin umbelliferone recorded a significantly less effect (3.05–22.4%), thus suggesting that the addition of the 3,3-dimethylallyl side chain may play a pivotal role in the mechanism of action underlying the observed effects, allowing 7-isopentenyloxycoumarin to selectively trigger a specific biomolecular target. When assayed on LoVo cells in the concentration range 0.01–100 μM, 7-isopentenyloxycoumarin showed a moderate growth inhibitory effect (IC_50_ = 30 μM), as pointed out in 2017 by Bisi and coworkers [27]. In the same study it was revealed that this phytochemical has a capacity to revert multidrug resistance equal to verapamil when assayed at a concentration of 2.5 μM in the same cancer cell line. One year later, Kafi and coworkers found that 7-isopentenyoxycoumarin was cytotoxic against K562 (human myelogeneous leukemia) cells in the concentration range 10–40 μM and that the same could modulate myeloid cell leukemia type-1 (Mcl-1) gene expression by hormesis in the same concentration range [28]. Finally, Marquez Duarte da Cruz and coworkers studied the mechanism of action of the in vitro and in vivo (Ehrlich ascites carcinoma in mice) anticancer activity of 7-isopentenyloxycoumarin [29]. First, its toxicity was evaluated: this coumarin did not provide an appreciable level of hemolysis on mice peripheral blood erythrocytes at a concentration of 2000 µg/mL and no deaths were observed in parallel in an in vivo acute non-clinical toxicity assay (LD_50_ = 1000 mg/kg). 7-Isopentenyloxycoumarin was also shown not to be genotoxic when assayed in the peripheral blood micronucleus test. No statistically significant differences were recorded with the control group of untreated cells. Accomplishing in vivo assays and applying a concentration range 25–50 mg/kg, 7-isopentenyloxycoumarin was able to reduce tumor volume and weight, both by about 50%. The same percentage of reduction was observed for the total viable cancer cells with respect to the controls. For what concerns the mechanism of action, it was found that the same led to a reduction of microvessel density around the tumor mass (24–58%) and, although only at the concentration of 50 mg/kg, to a decrease (45%) of the release of chemokine (C-C motif) ligand 2 (CCL2). Thus, it was hypothesized that 7-isopentenyloxycoumarin could reduce tumor growth and development by the inhibition of angiogenesis and CCL2 release in the tumor microenvironment.

Other major issues about 7-isopentenyloxycoumarin concern its neuroprotective properties, its effects on glucose metabolism, and its modulatory activity on melanogenesis.

### 3.2. Neuroprotective Activity

In 2016 Okuyama and coworkers investigated the ability of 7-isopentenyloxycoumarin to abolish the microglial activation and dopaminergic neuronal cell death in a lipopolysaccharide (LPS)-induced model of Parkinson’s disease [13]. This oxyprenylated coumarin, applied at a concentration of 100 μM in every experiment, was able to decrease (80.9%) the LPS-promoted hyperactivation of microglia in the central nervous system, while exhibited only a weak effect on the so called “reactive gliosis”, namely the activation of astrocytes by LPS. Furthermore, 7-isopentenyloxycoumarin showed a marked preventive effect against neuronal death of dopaminergic cells induced by the inflammatory process evoked by the administration of LPS (2.3-fold decrease in respect to controls), confirming what was observed in vitro and in vivo in 2008 and 2009 by Epifano and coworkers [12,30,31]. In 2010, Karimi and coworkers investigated the extent of inhibition of 7-isopentenyloxycoumarin on acetylcholinesterase [32]. Quite a low value to this respect was recorded (11.7% at a concentration of 100 μM). This was confirmed in 2019 by Erdogan Orhan and coworkers (12.5% inhibition at the same concentration value) [33]. However, in the latter investigation, it was found that 7-isopentenyloxycoumarin exerted a potent inhibitory activity on butyrylcholinesterase (86.9%, IC_50_ = 11.77 μM), about 3.5-fold higher than galanthamine, which was used as the reference drug (IC_50_ = 46.6 μM). The interaction of the coumarin with this enzyme has been also detailed by in silico experiments showing how 7-isopentenyloxycoumarin is able to tightly interact in its active site with π–π stacking interactions from its aromatic rings with Trp82, and also shows a water-mediated hydrogen bond interaction between the polar site of the ligand and Gly197.

### 3.3. Effects on Glucose Metabolism

Concerning the effect of 7-isopentenyloxycoumarin on glucose metabolism, in 2017 Genovese and coworkers investigated its effect on glucose transporter 4 (GLUT4) in L6 rat skeletal myoblasts [34]. When administered in the concentration range 0.1–10 μM, the oxyprenylated coumarin increased the glucose uptake by these cells in a dose-dependent manner. At the same time, 7-isopentenyloxycoumarin led to the translocation of GLUT4 to the plasma membrane. Both effects were very similar to those evoked by insulin (0.1 μM) used as the reference drug. It is noteworthy to underline how, also in this case, the unprenylated counterpart, umbelliferone, provided a very weak activity.

### 3.4. Stimulation of Melanogenesis

The last reports on the biological activity of 7-isopentenyloxycoumarin were in 2018 and 2019 by Fiorito and coworkers. The effects of this coumarin on the overall melanogenetic process in cultured Melan-a melanocytes were investigated [35,36]. On these cells, 7-isopentenyloxycoumarin showed no effect on cell viability even at high concentrations. At a dose of 40 μM, it showed a marked induction of melanin biosynthesis. This effect was later seen to be dose- and time-dependent with the highest values recorded after 72 h treatment (6-fold increase in respect to controls). Trying to depict a putative mechanism of action underlying the observed tanning activity, Fiorito and coworkers also showed that 7-isopentenyloxycoumarin was able to induce tyrosinase-related proteins 1 and 2 as well as microphthalmia-associated transcription factors. More interestingly, this coumarin was seen to tightly interact with the estrogen receptor β (ER-β) as an effective agonist. Thus, it was hypothesized that the massive increase in melanin biosynthesis could be mediated by the strong activation of this receptor. As a confirmation, the effect of pre-treatment of Melan-a cells with the ER-β antagonist Faslodex^®^ 1 μM was abolished by 7-isopentenyloxycoumarin, which induced a 2-fold increase in melanin biosynthesis in the same cell line with respect to controls. Furthermore, an involvement of the aryl hydrocarbon receptor could be hypothesized [8].

## 4. Conclusions

From the data reported in this miniature review, we have shown that a growing interest towards 7-isopentenyloxycoumarin has been recorded over the last decade with the characterization of novel plant sources and further insights into its biological activities and associated mechanisms of action. Thus, the idea that 7-isopentenyloxycoumarin may be regarded as a novel biologically active natural product with a great potential for the future has been enforced. It also has to be considered that this compound is nowadays very easy to synthesize in very high yields and could be available on the gram scale to perform further phytochemical and pharmacological studies, which surely deserve to be accomplished, to outline a more complete profile with designation of its biological targets, mechanism of action, and patterns of main and side effects. It is clear how 7-isopentenyloxycoumarin can nowadays effectively be regarded as an additional, although less abundant, component of the phytochemical pool of several plant species belonging to numerous families. The well-characterized biological effects so far assessed for this coumarin both in vitro and in vivo, in particular as an antitumor, cancer chemopreventive, and neuroprotective agent, reported herein do underline and enforce the role of the 7-isopentenyloxy-containing plants and their phytopreparations as beneficial means for human and animal health. In these respective “phytochemical contexts”, this coumarin may display a synergism of action with already reported phytochemicals such as flavonoids and phenolic acids with proven nutraceutical properties. Such a potential synergy surely deserves to be further detailed in the future, as does the search for other plant sources with a particular reference to food ones. Finally, it is noteworthy to highlight, discovered in the course of the new pharmacological investigations performed across the last decade, the importance of the biological activity in terms of structure activity relationship of the 3,3-dimethylallyl side chain, which confirms previously acquired data [37].

## Figures and Tables

**Figure 1 molecules-25-05923-f001:**
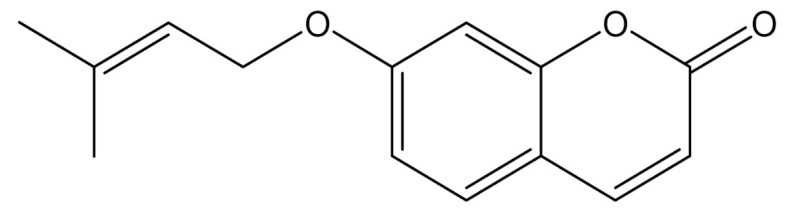
Structure of 7-isopentenyloxycoumarin.

**Figure 2 molecules-25-05923-f002:**
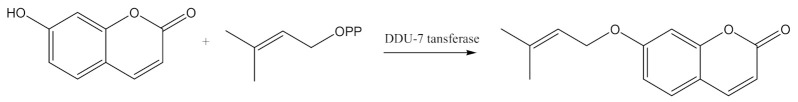
Biosynthetic pathway of 7-isopentenyloxycoumarin in *A. majus* cells.

**Table 1 molecules-25-05923-t001:** Natural sources of 7-isopentenyloxycoumarin (sources characterized during the last decade in bold).

Family	Source	Ref.
--	**Raw propolis**	[9]
Amaranthaceae	***Amaranthus retroflexus***	[12]
	***Spinacia oleracea***	[16]
Apiaceae	*Ammi majus*	[5]
	***Anethum graveolens***	[11]
	***Angelica archangelica***	[11]
	*Angelica ursina*	[5]
	*Heracleum dissectum*	[5]
	*Heracleum lanatum*	[5]
	*Lomatium nevadense*	[5]
	*Peucedanum stenocarpum*	[5]
	***Pimpinella anisum***	[11]
	*Scandix pectens-veneris*	[5]
	*Seseli libanotis*	[5]
	*Tordylium apulum*	[5]
Asteraceae	*Baccharis pedunculata*	[5]
	*Haplopappus deserticola*	[5]
	*Haplopappus multifolius*	[5]
	*Haplopappus tenuisectus*	[5]
	***Helianthus annuus***	[18]
	*Heterotheca inuloides*	[5]
	***Isocoma*** **spp.**	[16]
	*Melampodium divaricatum*	[5]
	*Ophryosporus angustifolius*	[5]
	*Tagetes florida*	[5]
	*Trichocline reptans*	[5]
Fabaceae	***Arachish ypogaea***	[18]
	***Glycine max***	[18]
Myrtaceae	***Melaleuca alternifolia***	[17]
Oleaceae	***Olea europea***	[18]
Poaceae	***Zea mays***	[18]
Rutaceae	*Asterolasia phebalioides*	[5]
	*Boenninghausenia albiflora*	[5]
	*Citrus limon*	[5]
	*Coleonema album*	[5]
	*Coleonema aspalathoides*	[5]
	*Diosma acmaeophylla*	[5]
	*Euodia vitiflora*	[5]
	*Haplophyllum patavinum*	[5]
	*Melicope hayesii*	[5]
	*Melicope semecarpifolia*	[5]
	*Melicope vitiflora*	[5]
	*Phebalium brachycalyx*	[5]
	*Ruta graveolens* L.	[5]
	*Zanthoxylum tingoassuiba*	[5]
Solanaceae	***Lycium barbarum***	[19]

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
