# Peer review of "7-Isopentenyloxycoumarin: What Is New across the Last Decade"

_molecules, 2020, doi:10.3390/molecules25245923_

Round 1

Reviewer 1 Report

Comments

This paper is a review article focused on 7-isopentenyloxycoumarin. The authors introduced recent years’ reports concerning this compound. Almost of the descriptions were seemed to appropriate. On the other hand, this reviewer has some asking and considers that it is necessary to clear noted below (1-9).

  1. lines 32, 60, and table 1, for example; The order of family name seemed to be inconsistent.

  1. Both “Asteraceae” and “Compositae” were wrote. Are they needed using separately?

  1. Concerning described family (ex. Apiaceae, Rosaceae, Asteraceae) plants including 7-isopentenyloxycoumarin, they are genetically close? Are there reports or considerations to chemotaxonomy?

  1. The size of oxygen and OPP are small comparing with the structures’ size in Figures 1 and 2. Please reconsider balance and check the stylistic rules of this journal.

  1. lines 86-96; The conclusion of the best method for extraction of 7-isopentenyloxycoumarin was established?

  1. lines 98-105; Do the 7-isopentenyloxycoumarin relate to the beneficial activities in mentioned here?

  1. line 128; Toxicities of these plants poisonous are concerned by 7-isopentenyloxycoumarin?

  1. line 154; Please change S-(+)- to italic S-.

  1. “Biological activity” section was difficult to read because the paragraph is long. It seemed to a list of reports, not review.

Author Response

First, all Authors wish to thank this reviewer for the effort anf time spent to revise our paper and to make valuable suggestions to get a better scientific quality. Please find below a point by point reply to the comments and suggestions raised

  1. lines 32, 60, and table 1, for example; The order of family name seemed to be inconsistent. FAMILIES HAVE BEEN IN ALPHABETIC ORDERS THROUGHOUT THE MANUSCRIPT

  1. Both “Asteraceae” and “Compositae” were wrote. Are they needed using separately? ONLY THE TERM "ASTERACEAE" WAS USED

  1. Concerning described family (ex. Apiaceae, Rosaceae, Asteraceae) plants including 7-isopentenyloxycoumarin, they are genetically close? Are there reports or considerations to chemotaxonomy? THERE ARE NO REPORTS TO THIS CONCERN IN THE LITERATURE SO WE COULD MENTION ANY RELEVANT DATA TO THIS REGARD IN OUR MANUSCRIPT

  1. The size of oxygen and OPP are small comparing with the structures’ size in Figures 1 and 2. Please reconsider balance and check the stylistic rules of this journal. THE SIZE OF FIGURES WAS ADJUSTED ACCORDINGLY

  1. lines 86-96; The conclusion of the best method for extraction of 7-isopentenyloxycoumarin was established? A SENTENCE STATING THIS CONCEPT HAS BEEN WRITTEN IN THE MANUSCRIPT

  1. lines 98-105; Do the 7-isopentenyloxycoumarin relate to the beneficial activities in mentioned here? A SENTENCE STATING THIS CONCEPT WITH SUPPORTING REFERENCES HAS BEEN WRITTEN IN THE MANUSCRIPT

  1. line 128; Toxicities of these plants poisonous are concerned by 7-isopentenyloxycoumarin? A SENTENCE STATING THIS CONCEPT HAS BEEN WRITTEN IN THE MANUSCRIPT

  1. line 154; Please change S-(+)- to italic S-. DONE

  1. “Biological activity” section was difficult to read because the paragraph is long. It seemed to a list of reports, not review. THIS PARAGRAPH WAS DIVIDED IN SUBSECTIONS

Reviewer 2 Report

This is a focused review on 7-isopentenyloxycoumarin by authors who have worked in the field and have acquired in depth knowledge. The manuscript is well organized and deserves publishing since it would be of interest to the audience of Molecules. One suggestion to the authors is to avoid the use of the term "anti-cancer" since in the strict pharmacognostical/pharmacological terminology it is reserved for compounds that have been tested in cancer patients and not for compounds only tested in cell lines (cytotoxic) or in animal models (anti-tumor).

Minor suggestions below:

  1. It would be advisable to avoid the phrases "the title natural product" and "the Authors" and be more definite.
  2. line 76. Use "syn" instead of "sin"
  3. line 117. A herb and not an herb.
  4. line 175. Delete "appeared"
  5. line 182. Delete "As the summary"
  6. lines 184-191 are unnecessary
  7. line 196. use "from" instead of "form"
  8. line 209. look at the syntax. "being" does not seem right
  9. line 212. The word "accomplished" is not appropriate. Use "performed: or re-write
  10. Give abbreviation for DAPI and all other abbreviations.
  11. line 241. "cells" instead of "cellas"
  12. line 242. "could" instead of "could and can".
  13. Be consistent with the past tense when describing past studies (lines 261, 270 etc)
  14. line 248. Maybe "in an in vivo ..."
  15. line 275. The correct is "butyrylcholinesterase"

Author Response

First, all Authors wish to thank this reviewer for the effort anf time spent to revise our paper and to make valuable suggestions to get a better scientific quality. Please find below a point by point reply to the comments and suggestions raised

One suggestion to the authors is to avoid the use of the term "anti-cancer" since in the strict pharmacognostical/pharmacological terminology it is reserved for compounds that have been tested in cancer patients and not for compounds only tested in cell lines (cytotoxic) or in animal models (anti-tumor). WE USED THE TERM CYTOTOXIC AND ANTI-TUMOR AS SUGGESTED

Minor suggestions below:

  1. It would be advisable to avoid the phrases "the title natural product" and "the Authors" and be more definite. BOTH TERMS HAVE BEEN DELETED THROUGHOUT THE WHOLE TEXT
  2. line 76. Use "syn" instead of "sin" DONE
  3. line 117. A herb and not an herb. DONE
  4. line 175. Delete "appeared" DONE
  5. line 182. Delete "As the summary" DONE
  6. lines 184-191 are unnecessary DELETED
  7. line 196. use "from" instead of "form" DONE
  8. line 209. look at the syntax. "being" does not seem right DONE
  9. line 212. The word "accomplished" is not appropriate. Use "performed: or re-write DONE
  10. Give abbreviation for DAPI and all other abbreviations. DONE
  11. line 241. "cells" instead of "cellas" DONE
  12. line 242. "could" instead of "could and can". DONE
  13. Be consistent with the past tense when describing past studies (lines 261, 270 etc) TIMES OF VERBS HAVE BEEN CORRECTED
  14. line 248. Maybe "in an in vivo ..." DONE
  15. line 275. The correct is "butyrylcholinesterase" DONE

Reviewer 3 Report

The paper describes a mini-review about new sources and bioactivities of 7-isopentenyloxycoumarin. The paper is interesting and should be published. However, some minor corrections are necessary on the text, as listed below:

Line 32: the correct name of the family is Thymelaeaceae.

Line 192: Table must be organized in alphabetical order by family and, subsequently, by genus.

Additionally, the same authors published at reference [5] a review about this compound covering the previous period. My suggestion is the table could be shown a summary of the occurrences of 7-isopentenyloxycoumarin covering the entire period.  The authors could highlight, for example, the new occurrences in bold.

Line 195: phenylpropanoids instead phenylopropanoids

Section new natural sources could be splitted in two topics: a) new sources and b) chromatographic methods for separation and quantification of the compound.

Line 199: Are there any methods in the literature that describe the synthesis of this compound? A topic on synthetic methods for obtaining the component may be interesting before describing biological activities.

Authors wrote in conclusion that this compound is nowadays very easy to synthesize in very high yields. Please describe the most important synthetic methods for obtaining this component.

Lines 200 to 304: I recommend the subdivision of topic 3 according to the biological activity.

Author Response

We thanks this reviewer for his/her effort in making valuable suggestions and corrections to our paper. All the comments has been addressed as outlined below

Line 32: the correct name of the family is Thymelaeaceae. CORRECTED

Line 192: Table must be organized in alphabetical order by family and, subsequently, by genus. DONE

Additionally, the same authors published at reference [5] a review about this compound covering the previous period. My suggestion is the table could be shown a summary of the occurrences of 7-isopentenyloxycoumarin covering the entire period. The authors could highlight, for example, the new occurrences in bold. THE TABLE WAS MODIFIED ACCORDINGLY

Line 195: phenylpropanoids instead phenylopropanoids DONE

Section new natural sources could be splitted in two topics: a) new sources and b) chromatographic methods for separation and quantification of the compound. THIS IS RATHER HARD TO CORRECT AS CHROMATOGRAPHIC METHODS ARE STRICTLY CONNECTED TO THE CHARACTERIZATION OF NOVEL SOURCES

Line 199: Are there any methods in the literature that describe the synthesis of this compound? A topic on synthetic methods for obtaining the component may be interesting before describing biological activities. THE SYNTHESIS WAS MENTIONED IN THE FIRST PART OF THE "NEW NATURAL SOURCES PARAGRAPH"

Authors wrote in conclusion that this compound is nowadays very easy to synthesize in very high yields. Please describe the most important synthetic methods for obtaining this component. VIDE SUPRA

Lines 200 to 304: I recommend the subdivision of topic 3 according to the biological activity. MODIFIED ACCORDINGLY

Reviewer 4 Report

The review paper present important information about 7-Isopentenyloxycoumarin. Authors focused on new sources of the compound and biological activities of it. In my opinion the paper is valuable and scientific soudness nevertheless have to be improve in terms of structure.

Please improve sections:  2. New natural sources and 3. Biological activity. The sections present significant information nevertheless should be divided into few subsection. Especially attention should be paid to bological activity due to this part should be enlarged and divided in terms of type of activity (i.e. anticancer, anti-nurodegerenation, etc) or/and type of studies: in vitro and in vivo.

Additionally, references should be unify and enriched with doi.

Author Response

We thank this referee for the effort and time spent to revise pour manuscript and provide valuable and precious suggestions to ameliorate its scientific quality. Below please find a point by point reply to the comments raised

Please improve sections: 2. New natural sources and 3. Biological activity. The sections present significant information nevertheless should be divided into few subsection. Especially attention should be paid to bological activity due to this part should be enlarged and divided in terms of type of activity (i.e. anticancer, anti-nurodegerenation, etc) or/and type of studies: in vitro and in vivo. DONE

Additionally, references should be unify and enriched with doi DONE

Round 2

Reviewer 4 Report

Dear Authors,

Thank you for the manuscript improvement. In this form, the paper is more readable and comprehensive.

Best regards